# Comparing the Clinical and Laboratory Features of Remitting Seronegative Symmetrical Synovitis with Pitting Edema and Seronegative Rheumatoid Arthritis

**DOI:** 10.3390/jcm10051116

**Published:** 2021-03-07

**Authors:** Misako Higashida-Konishi, Keisuke Izumi, Satoshi Hama, Hiroshi Takei, Hisaji Oshima, Yutaka Okano

**Affiliations:** 1Department of Connective Tissue Diseases, National Hospital Organization Tokyo Medical Center, Tokyo 1528902, Japan; izz@keio.jp (K.I.); shama@ntmc-hosp.jp (S.H.); htakei@ntmc-hosp.jp (H.T.); hoshimamac@mac.com (H.O.); yutakaokano@mac.com (Y.O.); 2Division of Rheumatology, Department of Internal Medicine, Keio University School of Medicine, Tokyo 1608582, Japan

**Keywords:** rheumatoid arthritis, synovitis, neoplasms, edema, inflammation

## Abstract

In seronegative arthritis with extremity edema, it is difficult to differentiate between remitting seronegative symmetrical synovitis with pitting edema syndrome (RS3PE) and seronegative rheumatoid arthritis (SNRA). We compared the clinical characteristics of RS3PE and SNRA in patients with and without malignancies. We retrospectively examined patients diagnosed with RS3PE (McCarty criteria) and SNRA at our hospital in 2007–2020. Malignancy was diagnosed within 2 years before or after RS3PE or SNRA diagnosis. Overall, 24 RS3PE and 124 SNRA patients were enrolled. The median ages were 79.5 and 68.5 years, and men comprised 54.2% and 37.1% of RS3PE and SNRA patients, respectively. RS3PE patients had higher inflammation levels (*p* = 0.004) and more incidences of malignancy (*p* = 0.034). Matching for age and sex, RS3PE patients had higher inflammation levels (*p* = 0.021) and more incidences of malignancy (*p* = 0.005). Overall, odds ratios (ORs) for malignancy were higher for older age (OR 1.06, *p* = 0.037), male sex (OR 4.34, *p* = 0.007), RS3PE patients (OR 4.83, *p* = 0.034), and patients with extremity edema (OR 4.83, *p* = 0.034). Inflammation levels and associated factors of malignancy were higher in RS3PE patients than in SNRA patients. Patients who are older, male, with extremity edema, or had RS3PE should be screened for malignancies.

## 1. Introduction

Remitting seronegative symmetrical synovitis with pitting edema (RS3PE) was first reported by McCarty et al. in 1985 [1]. It is characterized by pitting edema of the extremities, sudden onset of polyarthritis, seronegativity for rheumatoid factor (RF), excellent response to glucocorticoids, and the absence of radiologically evident erosions [1]. RS3PE mainly affects the joints of the extremities, especially the metacarpophalangeal (MCP) and proximal interphalangeal (PIP) phalanges, wrists, shoulders, elbows, knees, and ankles [2]. Although the pathophysiology of RS3PE remains unclear, vascular endothelial growth factor (VEGF) serum levels have been found to be elevated in patients with RS3PE [3]. The increase in vascular permeability by VEGF is thought to be responsible for the development of pitting edema of the dorsum of both hands and both feet in patients with RS3PE [3].

Initially, RS3PE was thought to be a type of older-onset rheumatoid arthritis (RA) [4] and was considered the same disease as seronegative RA and polymyalgia rheumatica (PMR) [5]. Subsequently, comparisons between PMR and RS3PE have been reported [6]. Kawashiri et al. reported the differences in musculoskeletal ultrasound findings of both hands between RS3PE and “seropositive” elderly onset RA; however, to our knowledge, no reports have compared the characteristics of RS3PE and “seronegative” RA [7].

RS3PE is often described as a paraneoplastic disease [8] and has been reported to have a high rate of malignancy development [9]. Paraneoplastic arthritis often presents as symmetrical polyarthritis, mainly affecting the wrist and fingers, and is often negative for RF and anti-cyclic citrullinated peptide antibody (ACPA) [10]. Early diagnosis of malignancy is clinically important because it improves survival. Therefore, examination for malignancy is necessary in such cases.

The primary aim of this study was to compare the clinical characteristics of RS3PE and seronegative RA and evaluate the frequency of concurrent malignancy. The secondary aim was to compare the clinical features with and without malignancies in patients with RS3PE and to compare the clinical features with and without malignancies in patients with seronegative RA.

## 2. Materials and Methods

### 2.1. Compliance with Ethical Standards

All procedures were performed in accordance with the ethical standards of the institutional and national research committees and the 1975/1983 Helsinki Declaration and its later amendments.

### 2.2. Study Design

This was a retrospective medical record study.

### 2.3. Patients

Medical records of consecutive patients diagnosed with RS3PE and seronegative RA at our hospital between 2007 and 2020 were retrospectively examined. Patients who were both ACPA- and RF-negative were included. Patients who met the criteria for both PMR and RS3PE were included in the RS3PE group and those who met the criteria for both PMR and seronegative RA were included in the seronegative RA group. PMR was diagnosed according to the 2012 European League Against Rheumatism/American College of Rheumatology (EULAR/ACR) Provisional Classification Criteria for PMR [11]. For patients diagnosed with PMR before 2012, we retrospectively reviewed whether they met the 2012 PMR classification criteria. Patients who met the criteria for both RA and RS3PE were diagnosed with RS3PE. However, those who had erosion were diagnosed with seronegative RA. We defined RS3PE and seronegative RA patients by excluding those who met the criteria for PMR as “pure RS3PE” and “pure seronegative RA.” Patients who met the criteria for both RA and PMR were diagnosed with seronegative RA. Patients with paraneoplastic polyarthritis were excluded from the group of patients with RS3PE or seronegative RA. Those with distal joint swelling that rapidly disappeared after tumor resection were diagnosed with paraneoplastic polyarthritis.

### 2.4. RS3PE Diagnosis

Patients were diagnosed with RS3PE when they met the McCarty et al. criteria [1]: (1) pitting edema of the dorsum of both hands and both feet, (2) sudden onset of polyarthritis, (3) seronegative for RF, and (4) no development of radiologically evident erosions.

### 2.5. Seronegative RA Diagnosis

Seronegative RA was diagnosed according to the 2010 EULAR/ACR criteria [12]. Patients who were first diagnosed with RS3PE or PMR and later diagnosed with seronegative RA were included in the seronegative RA group.

### 2.6. Clinical and Laboratory Features

We examined the affected joints and evaluated them for systemic signs and symptoms (temperature ≥38.0 °C, malaise or fatigue, weight loss, morning stiffness lasting at least 1 h, and edema). The affected joints were the shoulders, elbows, wrists, fingers (MCP and interphalangeal (IP)/PIP joints), hips, knees, ankles, and toes (MCP and IP/PIP joints). Edema was evaluated separately as edema of only hands, only feet, and of both limbs. We also measured the erythrocyte sedimentation rate (ESR) and the levels of C-reactive protein (CRP), hemoglobin (Hb), albumin (Alb), lactate dehydrogenase (LDH), and matrix metalloproteinase 3 (MMP-3). Smokers were defined as those who had a smoking history within 2 years before and after RS3PE or seronegative RA diagnosis. If there were evaluable examinations, ultrasound imaging, breast imaging, joint X-ray imaging, chest computed tomography (CT), abdominal CT, pelvic CT, positron emission tomography/CT, joint magnetic resonance imaging, upper and lower gastrointestinal endoscopy, gynecological examination, and pathological tests were performed.

### 2.7. Statistical Analysis

The first analysis was performed on clinical and laboratory features of patients with RS3PE and seronegative RA. The secondary analysis was performed on the above evaluations with a 1:2 matching for age and sex. All data were analyzed using JMP version 14.0 (SAS Institute, Cary, NC, USA). The third analysis was performed to compare the clinical features of patients with or without malignancy among patients with RS3PE or seronegative RA. Univariate analysis, Fisher’s exact test, and logistic regression analysis were applied to evaluate the associated factor of malignancy. A probability level less than 0.05 was used as the criterion of significance. Results that did not follow the Gaussian distribution were expressed as the median of the 25–75th percentile (interquartile range), and results that followed the Gaussian distribution were expressed as mean ± standard deviation. The odds ratio (OR) and its 95% confidence interval (95% CI) indicated the increased or decreased risk of malignancy associated with a one-unit change in the predictor variable for continuous variables. For dichotomous variables, the OR indicated the risk of malignancy associated with the presence of the feature compared to the absence of the characteristic. In the case of missing data, the number of patients with available data was specified.

## 3. Results

We enrolled 24 consecutive patients with RS3PE examined at our hospital between 2007 and 2020 (Appendix A). Initially, 25 patients were diagnosed with RS3PE according to the criteria of McCarty et al. [1]. However, one patient was later diagnosed with paraneoplastic polyarthritis with rapid remission of distal swelling with pitting edema after tumor resection and was excluded from the RS3PE group. Only one patient was diagnosed with paraneoplastic polyarthritis: an 81-year-old woman who presented with polyarthritis and edema of both hands and feet. Her blood test showed high levels of CRP (2.2 mg/dL). During examination, she was diagnosed with cancer of the pancreatic body and underwent surgery to remove the body and tail of the pancreas. The postoperative course is uneventful. One month after the operation, the polyarthritis resolved and the levels of CRP decreased (0.1 mg/dL) without the use of medication.

In the control group, 124 consecutive patients with seronegative RA during the same period were enrolled. Appendix A shows the patient diagnosis flow.

Figure 1 shows the breakdown of patients according to the criteria for RS3PE, RA, and PMR. The RS3PE group consisted of Group A, B, and C patients. The seronegative RA group consisted of Group D and E patients. In the RS3PE and seronegative RA groups, two and 17 patients, respectively, met the 2012 EULAR/ACR provisional criteria for PMR [11] (Figure 1). After excluding those patients, 22 patients (Groups A and B, Figure 1) with RS3PE and 107 patients (Group D, Figure 1) with seronegative RA were analyzed with similar results to those obtained at baseline, including the incidence of comorbid malignancies (Appendix A).

### 3.1. Comparison of Clinical and Laboratory Features of RS3PE and Seronegative RA

In the first analysis, baseline characteristics at diagnosis of the 24 RS3PE patients were compared with those of the 124 seronegative RA patients (Table 1). The onset age of RS3PE was significantly higher than that of seronegative RA. The RS3PE patients had less swollen small joints and significantly higher levels of CRP, LDH, and MMP-3 than the seronegative RA patients. The numbers of swollen and/or tender joints were similar in both groups, except for the elbows and fingers, which were more affected in the seronegative RA patients. The ankles were more affected in the RS3PE patients than in the seronegative RA patients.

Malignancies were detected in six of 24 (25%) patients in the RS3PE group and in eight of 124 (6.5%) patients in the seronegative RA group within 2 years before and after RS3PE/seronegative RA diagnosis. The malignancy incidence rate in the RS3PE group was significantly higher than that in the seronegative RA group (*p* = 0.034). Table 2 presents the patients with malignancies and the types of malignancies. Advanced malignancies were not found in the RS3PE patients. There was one case of advanced malignancy (pancreatic cancer) in a seronegative RA patient.

### 3.2. Comparison of Clinical and Laboratory Features of RS3PE and Seronegative RA with a 1:2 Matching for Age and Sex

Since the incidence of malignancies depends on age and sex, we performed a 1:2 matching in the second analysis. After matching for age and sex, 24 patients with RS3PE and 48 with seronegative RA were selected for comparison. Malignancies were significantly more common in the RS3PE than in the seronegative RA patients (Table 3). The RS3PE patients had less swollen and tender joints and significantly higher CRP levels than the seronegative RA patients.

### 3.3. Comparison of Clinical Features of Patients with and without Malignancies among the RS3PE and Seronegative RA Patients

Table 4 shows a comparison of the clinical features of the patients with and without malignancies. There were 14 patients with malignancies and 134 patients without malignancies, with median ages of 79.5 and 69.5 years, respectively (*p* = 0.032). Furthermore, 71.4% and 36.6% of the patients, respectively, were men (*p* = 0.011). The RS3PE patients constituted 42.9% and 13.4% (*p* = 0.034) of the patients with and without malignancies, respectively. Patients with malignancies had more edema of both hands and both feet (*p* = 0.034) than those without malignancies. There was no difference between the groups in terms of percentage of patients who fulfilled the criteria for PMR (*p* = 1.00). In terms of overall ORs for malignant comorbidities among the patients with RS3PE or seronegative RA, older age (OR 1.06, 95% CI 1.002–1.11, *p* = 0.037), male sex (OR 4.34, 95% CI 1.29–14.57, *p* = 0.007), RS3PE (OR 4.83, 95% CI 1.50–15.56, *p* = 0.034), and edema of both hands and both feet (OR 4.83, 95% CI 1.50–15.56, *p* = 0.034) were associated with the presence of comorbid malignancies. Seronegative RA (OR 0.21, 95% CI 0.06–0.07, *p* = 0.034) and increased Hb levels in men (OR 0.51, 95% CI 0.33–0.81, *p* = 0.005) were associated with the absence of comorbid malignancies (Table 5).

### 3.4. Comparison of Baseline Characteristics between RS3PE Patients with and without Malignancies

No clinical differences were noted between the RS3PE patients with and without malignancies (Appendix A).

### 3.5. Comparison of Baseline Characteristics between Seronegative RA Patients with and without Malignancies

The seronegative RA patients with malignancies had less swollen large joints (*p* = 0.027), lower MMP-3 levels (83.8 vs. 173.0 ng/mL, *p* = 0.07), lower ESRs in women (19.0 vs. 55.0 mm/h, *p* = 0.020), and higher Hb levels in women (13.7 ± 1.3 vs. 11.6 ± 1.8, *p* = 0.045) than those without malignancies (Appendix A).

## 4. Discussion

### 4.1. Comparison of Clinical and Laboratory Features of RS3PE and Seronegative RA

We found that patients with RS3PE were characterized by an older age at onset, higher affectation of the ankles compared to the elbows and fingers, higher levels of CRP and ESR, and a higher malignancy rate compared to patients with seronegative RA. These results (Table 1) are similar to those of Olive et al. [2], who reported that, in RS3PE patients, the MCP (81.5%) and PIP joints (70.4%), wrists (55.5%), shoulders (48%), knees (33.3%), and ankles (25.9%) were more frequently affected, while the elbows (11.1%) were less frequently affected. Patients with RS3PE had swollen and/or tender finger joints less frequently than those with seronegative RA (79.2% vs. 96.8%, *p* = 0.022). The reason for this is that patients with seronegative RA must present with 11 or more swollen or tender joints, including at least one small joint, to meet the 2010 EULAR/ACR criteria for RA [12]. This suggests that patients with seronegative RA tend to have many small joints affected. In our study, RS3PE more frequently affected the joints of the ankles than did seronegative RA. The high incidence of affected joints of the ankles in RS3PE patients may be due to attending physicians determining swelling in the ankle because of lower extremity edema in RS3PE patients.

The number of affected joints in the RS3PE patients was lower than that in the seronegative RA patients; however, the levels of CRP, and MMP-3 were higher. When analyzed with a 1:2 matching for age and sex, CRP levels were higher in the RS3PE group than in the seronegative RA group, while MMP-3 levels were comparable between the groups (Table 3). This implies that RS3PE and seronegative RA are essentially different diseases. Patients with RS3PE have often been reported to be positive for human leukocyte antigen (HLA)-B7, -Cw7, and -DQw2 [13], but not for HLA-DRB1, which is positive in RA [13,14]. Furthermore, RS3PE patients have higher levels of VEGF than RA patients [3]. This suggests that the pathogenesis of RS3PE is different from that of seronegative RA. Malignancies such as advanced cancers [15] and kidney cancers [16], which cause high levels of CRP, were not found in the RS3PE patients in this study.

PMR and seronegative RA have both positive HLA-DRB1, which may suggest that their etiologies may be the same; however, there are differences regarding their clinical manifestations. In PMR patients, there is significantly more frequent bilateral shoulder and hip pain and significantly less frequent peripheral arthritis (peripheral synovitis) than in RA patients [11]. Based on the distribution of the affected joints, it is not difficult to distinguish PMR from seronegative RA. Therefore, when the primary symptom of a patient who meets the criteria for PMR is peripheral arthritis; a diagnosis of RA is often made when the patient also meets the criteria for RA.

Compared to RS3PE, PMR has also been found to be significantly more common in male patients with a higher frequency of hip morning stiffness and pain [6]. Salvarani et al. [17] reported 19 cases of PMR with distal extremity swelling with pitting edema. However, edema in both hands and both feet was present in only three of the 19 cases, and all three cases met the criteria for RS3PE [1], although there are some missing data on RF. PMR with distal extremity swelling with pitting edema appears to identify a more benign disease subset than PMR without edema [18]. Patients who met the criteria for both PMR and RS3PE have previously been categorized as RS3PE [6,19]. Therefore, PMR with edema in all extremities could have been defined as RS3PE.

In our study, the patients who met the criteria for both RS3PE and PMR were defined as having RS3PE, and those who met the criteria for both seronegative RA and PMR were defined as having seronegative RA. Two (8.3%) and 17 (13.7%) patients with RS3PE and seronegative RA met the criteria for PMR [11], respectively. Excluding these patients who met the criteria for PMR, we reanalyzed 22 “pure RS3PE” and 107 “pure seronegative RA” patients. There were no differences in clinical characteristics and results between the “pure RS3PE” and “pure seronegative RA” groups, including the incidence of comorbid malignancies. These results suggest that it is not possible to differentiate RS3PE from seronegative RA regardless of the patients meeting the criteria for PMR. In paraneoplastic syndromes in rheumatology, musculoskeletal symptoms are known to occur in the joints and muscles [20] and PMR-like symptoms are also known to develop [21]. In our study, however, there was no relationship between meeting the PMR criteria and the presence or absence of malignancies (Table 5).

### 4.2. Comparison between RS3PE/Seronegative RA with and without Malignancies

Comorbid malignancies were found in 25.0% and 6.5% of the RS3PE and seronegative RA patients, respectively (Table 1). Based on data from the National Cancer Institute of Japan [22], the 4-year incidences of malignancies (2 years before and after the diagnosis of RS3PE/seronegative RA) in the Japanese population of the same age were 9.1% and 6.3% in RS3PE and seronegative RA patients, respectively. Thus, compared with the Japanese population, the incidence of comorbid malignancies was higher in the RS3PE group and comparable in the seronegative RA group. This is consistent with the findings of a previous report that the incidence of malignancies is higher in patients with RS3PE than in the general population [9]. The types of malignancies associated with RS3PE [23] include stomach, rectal, and prostate cancers, as observed in our study.

### 4.3. Comparison between RS3PE Patients with and without Malignancies

In the current study, there was no significant difference in the clinical characteristics of RS3PE between patients with and without malignancies (Appendix A). Origuchi et al. reported that RS3PE with malignancies has higher MMP-3 serum levels than RS3PE without malignancies, due to the abundant production of MMP-3 owing to malignancies [24]. In our study, there was no difference in MMP-3 levels. This discrepancy may have been due to the small number of cases both in the study by Origuchi et al. [24] and ours. These authors included eight patients with malignancy out of a total of 33 patients with RS3PE, and our study included six patients with malignancy out of a total of 24 patients with RS3PE. Due to the small number of cases to be analyzed, sufficient detection power may not have been obtained. These authors also included not only patients with edema of the hands and feet, but also that of only hands or only feet, which is different from our inclusion criteria that included patients with edema in both hands and both feet, similar to the study of McCarty et al. [1]. There was no difference in MMP-3 levels when analyzed separately by sex.

### 4.4. Comparison between Seronegative RA Patients with and without Malignancies

In our study, the seronegative RA patients with malignancies had lower MMP-3 levels and fewer swollen large joints than those without malignancies. Although MMP-3 serum levels can be elevated with steroids [25], all patients in this study had not used steroids before seronegative RA diagnosis. Additionally, patients with malignancies had fewer swollen large joints than those without malignancies (Appendix A). Serum levels of MMP-3 have been reported to be higher in RA patients with synovitis in large joints [26]. The MMP-3 serum levels did not correlate with the number of tender and swollen joints used in the core set of ACR, but they correlated with the Lansbury’s joint scores, which have a high coefficient for large joints [27]. Therefore, in our study, the low circulating levels of MMP-3 in seronegative RA patients with malignancy may be due to the small number of swollen large joints.

### 4.5. Comparison between Seronegative RA and RS3PE Patients with and without Malignancies

We also examined the differences in the clinical characteristics of the overall patients with and without malignant comorbidities. The ORs of the patients with malignancies were higher for older age, male sex, RS3PE, and edema of both hands and both feet (Table 5). Regarding older and male patients, these results are consistent with data from the National Cancer Institute of Japan and the general Japanese trend. The high ORs of RS3PE and edema in both hands and both feet for malignancy also suggest that a thorough examination for malignancies should be performed in patients with RS3PE.

### 4.6. Limitations

Our study has several limitations. First, this was a retrospective study. Therefore, we employed matching to minimize selection bias. Second, 23 seronegative RA patients (one with malignancy, 22 without malignancies) and eight RS3PE patients (three with malignancies, five without malignancies) could not be followed for ≥2 years after the diagnosis of seronegative RA and RS3PE, respectively. Nevertheless, the results were not different after the exclusion of these patients. In our study, the incidence of malignancies was defined within 2 years before and after RS3PE or seronegative RA diagnosis; however, it is not clear within what year malignancy should be included. Some reports included comorbid malignancies within a definite period after the onset of RS3PE [6,24], while other reports did not present a definite period [9,28]. The significant difference in the incidence of comorbid malignancies between the RS3PE and seronegative RA groups was noted even when including malignancies within 1 or 3 years before or after the diagnosis of RS3PE/RA. Third, our study population was small. Since RS3PE is a rare disease and this was a single center study, multicenter validation studies are warranted. Finally, there were some missing data on Alb and MMP-3, but there were no missing data on important indices such as CRP and ESR.

## 5. Conclusions

Patients with RS3PE had higher CRP levels and a higher risk for malignancy than those with seronegative RA. As RS3PE patients are likely to have malignancies, it is necessary to thoroughly examine for malignancies at RS3PE diagnosis.

The seronegative RA patients with malignancies had lower MMP-3 levels and fewer swollen large joints at RA diagnosis than those without malignancy. Furthermore, among seronegative RA patients, it is recommended that patients with lower MMP-3 levels and fewer swollen large joints should be screened for malignancy.

These findings may enable the performance of a differential diagnosis between RS3PE and seronegative RA. Moreover, this may encourage clinicians to examine for malignancies in patients with RS3PE, contributing to improved patient outcomes.

## Figures and Tables

**Figure 1 jcm-10-01116-f001:**
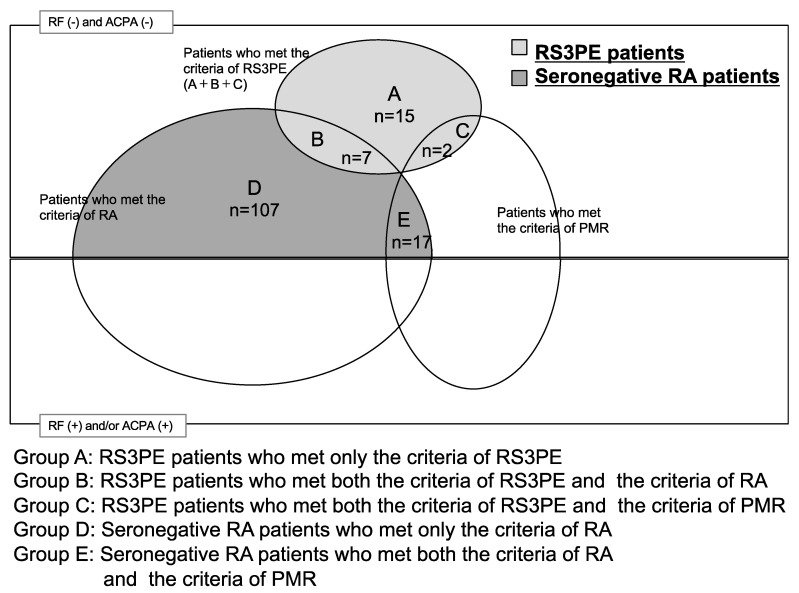
Diagnostic criteria for RS3PE and seronegative RA, as used in this study. Patients in Group A met only the criteria for RS3PE. Patients in Group B met the criteria for both RS3PE and RA. Patients in Group C met the criteria for both RS3PE and PMR. Patients in Group D met only the criteria for RA. Patients in Group E met the criteria for both RA and PMR. The RS3PE group consisted of Group A + B + C patients. The seronegative RA group consisted of Group D + E patients. No patients met the criteria for RS3PE, RA, and PMR. ACPA, anti-cyclic citrullinated peptide antibody; PMR, polymyalgia rheumatica; RA, rheumatoid arthritis; RF, rheumatoid factor; RS3PE, remitting seronegative symmetrical synovitis.

**Table 1 jcm-10-01116-t001:** Patient baseline characteristics at diagnosis.

Characteristics	RS3PE Patients (*n* = 24)		Seronegative RA Patients (*n* = 124)		*p* Value
Age, median (IQR), years	79.5 (73.8–86.5)		68.5 (58.5–78.0)		<0.001
Length of follow-up, median (IQR), months	31.5 (12.0–109.0)		62.9 (30.7–98.4)		0.09
Male sex, *n* (%)	13 (54.2)		46 (37.1)		0.17
Smoking, *n* (%)	5 (20.8)		23 (18.6)		0.78
Diabetes, *n* (%)	6 (25.0)		14 (11.3)		0.10
Hypertension, *n* (%)	12 (50.0)		41 (33.1)		0.16
Hyperlipidemia, *n* (%)	5 (20.8)		33 (26.1)		0.62
Swollen or/and tender joints, *n* (%)					
Shoulders	8 (33.3)		67 (54.3)		0.08
Elbows	2 (8.3)		53 (42.7)		0.001
Wrists	17 (70.8)		100 (80.7)		0.28
Fingers	19 (79.2)		120 (96.8)		0.022
Hips	4 (16.7)		13 (10.5)		0.48
Knees	9 (37.5)		59 (47.6)		0.38
Ankles	18 (75.0)		65 (52.4)		0.046
Toes	8 (33.3)		35 (28.2)		0.63
Patients with swollen large joints, *n* (%)	17 (70.8)		64 (51.6)		0.12
Patients with swollen small joints, *n* (%)	21 (87.5)		124 (100.0)		0.024
Number of swollen large joints, median (IQR), *n*	2.0 (0.0–2.8)		1.0 (0.0–2.0)		0.17
Number of swollen small joints, median (IQR), *n*	3.0 (1.3–13.3)		9.0 (5.0–15.0)		0.33
28 swollen joints, median (IQR), *n*	4.0 (1.3–10.8)		8.0 (5.0–14.0)		0.29
28 tender joints, median (IQR), *n*	6.5 (4.3–12.0)		11.0 (7.3–15.0)		0.15
Patients with erosion, *n* (%)	0 (0.0)		39 (31.5)		<0.001
Systemic signs and symptoms, *n* (%)					
Temperature ≥38 °C	2 (8.3)		7 (5.7)		0.64
Malaise or fatigue	3 (12.5)		8 (6.5)		0.39
Weight loss	5 (20.8)		12 (9.7)		0.16
Morning stiffness (lasting at least 1 h)	2 (8.3)		31 (25.0)		0.11
Edema (both hands and feet)	24 (100.0)		0(0)		<0.001
Edema (only hands)	0 (0.0)		1 (0.8)		1.0
Edema (only feet)	0 (0.0)		19 (15.3)		<0.001
CRP, median (IQR), mg/dL	8.2 (4.0–14)		2.8 (0.7–6.6)		0.004
ESR, median (IQR), mm/h					
Men+Women	91.0 (59–112.5)		55.0 (32.0–90.0)		0.07
Men	85.0 (28.5–114.5)		57.0 (31.0–90.0)		0.36
Women	91.0 (82–113)		54.0 (32.0–88.0)		0.010
Alb, median (IQR), g/dL	3.5 (3.0–3.7)	(*n* = 23) *	3.9 (3.4–4.1)	(*n* = 100) *	0.012
LDH, median (IQR), U/L	197.0 (161–234)		176.0 (155.5–195)		0.07
MMP-3, median (IQR), ng/mL					
Men+Women	378.5(243.3–662.2)	(*n* = 16) *	162.0 (82.2–401.1)	(*n* = 115) *	0.022
Men	359.4(269.1–435.4)	(*n* = 7) *	211.0 (115.3–420.9)	(*n* = 45) *	0.08
Women	414.1(92.8–997.2)	(*n* = 9) *	151.0 (47.2–348.5)	(*n* = 70) *	0.07
Hb, mean ± SD, g/dL					
Men + Women	10.7 ± 1.8		11.9 ± 1.8		0.024
Men	10.8 ± 2.0		12.2 ± 1.7		0.10
Women	10.6 ± 1.5		11.7 ± 1.8		0.12
Malignancy (within 2 years before and after the diagnosis of RS3PE or seronegative RA), *n* (%)	6 (25.0)		8 (6.5)		0.034
Patients fulfilling the classification criteria for RA [11,12], *n* (%)	7 (29.2)		124 (100.0)		<0.001
Patients fulfilling the classification criteria for PMR [10], *n* (%)	2 (8.3)		17 (13.7)		0.74
Patients fulfilling the classification criteria for RA [11,12] + PMR [10], *n* (%)	0 (0.0)		17 (13.7)		0.08

Alb, albumin; CRP, C-reactive protein; ESR, erythrocyte sedimentation rate; Hb, hemoglobin; IQR, inter quartile range; LDH, lactate dehydrogenase; MMP-3, matrix metalloproteinase 3; PMR, polymyalgia rheumatica; RA, rheumatoid arthritis; RS3PE, remitting seronegative symmetrical synovitis with pitting edema; SD, standard deviation. * In the case of missing data, the number of patients with available data was specified.

**Table 2 jcm-10-01116-t002:** Patients with malignancies 2 years before and after RS3PE or seronegative RA diagnosis.

Sex, Age (years)	Interval between Diagnosis of RS3PE/Seronegative RA and Malignancies (Months)	Malignancy Type
RS3PE		
M, 81	–24	Prostate cancer
M, 78	–24	Prostate cancer
M, 78	–11	Rectal cancer
F, 87	0 (+5 days)	Pancreatic cancer
M, 79	0 (+6 days)	Stomach cancer
M, 80	3	Rectal cancer
**Seronegative RA**		
M, 84	–20	Rectal cancer
F, 64	–17	Uterine cancer
M, 82	–6	Ascending colon cancer
M, 69	–5	Small cell lung cancer
F, 58	–4	Breast cancer
F, 80	1	Breast cancer
M, 67	9	Diffuse large B cell lymphoma
M, 83	18	Pancreatic cancer

RA, rheumatoid arthritis; RS3PE, remitting seronegative symmetrical synovitis with pitting edema; M, male; F, female.

**Table 3 jcm-10-01116-t003:** Baseline characteristics at diagnosis of RS3PE and seronegative RA patients with a 1:2 matching for age and sex.

Characteristic	RS3PE Patients (*n* = 24)	Seronegative RA Patients (*n* = 48)	*p* Value
Age, median (IQR), years	79.5 (73.8–86.5)	79.5 (73.3–85.3)	0.58
Male sex, *n* (%)	13 (54.2)	23 (47.9)	0.80
Swollen or/and tender joint, *n* (%)			
Shoulders	8 (33.3)	33 (68.8)	0.006
Elbows	2 (8.3)	19 (39.6)	0.006
Wrists	17 (70.8)	42 (87.5)	0.11
Fingers	19 (79.2)	46 (95.8)	0.037
Hips	4 (16.7)	6 (12.5)	0.72
Knees	9 (37.5)	19 (39.6)	1.00
Ankles	18 (75.0)	25 (52.1)	0.08
Toes	8 (33.3)	11 (22.9)	0.40
Patients with swollen large joints, *n* (%)	17 (70.8)	26 (54.2)	0.21
Patients with swollen small joints, *n* (%)	21 (87.5)	48 (100.0)	0.034
Number of swollen small joints, median (IQR), *n*	3.0 (1.3–13.3)	9.0 (6.0–15.0)	0.021
28 swollen joints, median (IQR), *n*	4.0 (1.3–10.8)	9.5 (6.0–15.0)	0.008
28 tender joints, median (IQR), *n*	6.5 (4.3–12.0)	11.0 (8.3–15.0)	0.019
Patients with erosion, *n* (%)	0 (0.0)	15 (31.3)	0.001
CRP, median (IQR), mg/dL	8.2 (4.0–14)	4.4 (1.3–8.4)	0.021
ESR, median (IQR), mm/h	91.0 (59–112.5)	75.0 (37.0–103.0)	0.25
LDH, median (IQR), U/L	197.0 (161–234)	184.5 (164.0–210.5)	0.26
MMP-3, median (IQR), ng/mL	378.5(243.3–662.2)	251.0 (124.0–555.0)	0.27
Hb, mean±SD, mg/dL	10.7 ± 1.8	11.5 ± 2.0	0.08
Malignancy (within 2 years before and after the diagnosis of RS3PE or seronegative RA), *n* (%)	6 (25.0)	1 (2.1)	0.005
Patients fulfilling the classification criteria for RA [11,12], *n* (%)	7 (29.2)	48 (100.0)	0.09
Patients fulfilling the classification criteria for PMR [10], *n* (%)	2 (8.3)	7 (14.6)	0.71
Patients fulfilling the classification criteria for RA [11,12] + PMR [10], *n* (%)	0 (0.0)	7 (14.6)	0.09

CRP, C-reactive protein; ESR, erythrocyte sedimentation rate; Hb, hemoglobin; IQR, interquartile range; LDH, lactate dehydrogenase; MMP-3, matrix metalloproteinase 3; PMR, polymyalgia rheumatica; RA, rheumatoid arthritis; RS3PE, remitting seronegative symmetrical synovitis with pitting edema; SD, standard deviation.

**Table 4 jcm-10-01116-t004:** Patient baseline characteristics at diagnosis of RS3PE and seronegative RA patients with or without malignancies.

Characteristics	With Malignancy (*n* = 14)		Without Malignancy (*n* = 134)		*p* Value
Age, median (IQR), years	79.5 (68.5–82.3)		69.5 (60.0–79.0)		0.032
Length of follow-up, median (IQR), months	40.6 (7.9–87.7)		57.4 (27.4–97.7)		0.36
Male sex, *n* (%)	10 (71.4)		49 (36.6)		0.011
Smoking, *n* (%)	5 (35.7)		23 (17.2)		0.14
Diabetes, *n* (%)	4 (28.6)		16 (11.9)		0.10
Hypertension, *n* (%)	5 (35.7)		48 (35.8)		1.00
Hyperlipidemia, *n* (%)	4 (28.6)		34 (25.4)		0.76
Swollen or/and tender joints, *n* (%)					
Shoulders	5 (35.7)		70 (52.2)		0.27
Elbows	5 (35.7)		50 (37.3)		1.00
Wrists	11 (78.6)		106 (79.1)		1.00
Fingers	13 (92.9)		126 (94.0)		1.00
Hips	2 (14.3)		15 (11.2)		0.67
Knees	5 (35.7)		63 (47.0)		0.56
Ankles	8 (57.1)		75 (56.0)		1.00
Toes	4 (28.6)		39 (29.1)		1.00
Patients with swollen large joints, *n* (%)	6 (42.9)		75 (56.0)		0.41
Patients with swollen small joints, *n* (%)	13 (92.9)		132 (98.0)		0.26
Number of swollen large joints, median (IQR), *n*	0.0 (0.0–2.3)		1.0 (0.0–2.0)		0.44
Number of swollen small joints, median (IQR), *n*	12.5 (4.3–18.5)		8.0 (4.0–13.0)		0.46
28 swollen joints, median (IQR), *n*	9.5 (3.5–16.8)		8.0 (4.0–12.0)		0.62
28 tender joints, median (IQR), *n*	7.5 (5.8–19.3)		10.0 (7.0–14.3)		0.74
Patients with erosion, *n* (%)	5 (35.7)		34 (25.4)		0.52
Systemic signs and symptoms, *n* (%)					
Temperature ≥ 38 °C	0 (0.0)		9 (6.7)		1.00
Malaise or fatigue	2 (14.3)		9 (6.7)		0.28
Weight loss	1 (7.1)		16 (12.0)		1.00
Morning stiffness (lasting at least 1 h)	4 (28.6)		29 (21.7)		0.55
Edema (both hands and feet)	6 (42.9)		18 (13.4)		0.034
Edema (only hands)	0 (0.0)		1 (0.8)		1.00
Edema (only feet)	0 (0.0)		19 (14.2)		0.22
CRP, median (IQR), mg/dL	6.1 (3.1–11.9)		3.1 (0.8–7.2)		0.08
ESR, median (IQR), mm/h					
Men + Women	46.0 (21.5–112.0)		59.0 (33.0–91.5)		0.88
Men	90.0 (35.0–114.0)		59.0 (31.0–90.5)		0.53
Women	22.5 (13.8–91.3)		59.0 (33.5–93.5)		0.15
Alb, median (IQR), g/dL	3.5 (3.1–4.0)		3.8 (3.3–4.1)	(*n* = 109) *	0.24
LDH, median (IQR), U/L	174.5 (166.8–214.8)		178.0 (155.0–206.3)		0.79
MMP-3, median (IQR), ng/mL					
Men+Women	220.0 (43.8–364.8)	(*n* = 13) *	181.0 (84.8–428.5)	(*n* = 118) *	0.75
Men	234.7 (133.0–364.8)	(*n* = 9) *	213.0 (116.0–426.2)	(*n* = 43) *	0.85
Women	37.7 (28.1–463.8)	(*n* = 4) *	162.0 (66.8–465.0)	(*n* = 75) *	0.13
Hb, mean ± SD, g/dL					
Men + Women	10.9 ± 2.0		11.8 ± 1.8		0.10
Men	10.3 ± 1.4		12.3 ± 1.7		0.001
Women	12.7 ± 2.3		11.5 ± 1.8		0.24
Patients diagnosed with RS3PE, *n* (%)	6 (42.9)		18 (13.4)		0.034
Patients diagnosed with RA [11,12], *n* (%)	8 (57.1)		116 (86.6)		0.034
Patients fulfilling the classification criteria for RA [11,12], *n* (%)	10 (71.4)		121 (90.0)		0.058
Patients fulfilling the classification criteria for PMR [10], *n* (%)	1 (7.1)		18 (13.4)		1.00

Alb, albumin; CRP, C-reactive protein; ESR, erythrocyte sedimentation rate; Hb, hemoglobin; IQR, interquartile range; LDH, lactate dehydrogenase; MMP-3, matrix metalloproteinase 3; RA, rheumatoid arthritis; RS3PE, remitting seronegative symmetrical synovitis with pitting edema; SD, standard deviation. * In the case of missing data, the number of patients with available data was specified.

**Table 5 jcm-10-01116-t005:** Risk factors for malignancy in patients with RS3PE or seronegative RA analyzed by univariate logistic regression analysis.

Characteristics	Odds Ratio	95% Confidence Interval	*p* Value
Age	1.06	1.002–1.11	0.037
Length of follow–up	0.999801	0.9994–1.0002	0.36
Male sex	4.34	1.29–14.57	0.007
Smoking	2.68	0.82–8.74	0.10
Diabetes	2.95	0.83–10.52	0.10
Hypertension	0.995	0.32–3.14	0.99
Hyperlipidemia	1.18	0.35–3.997	0.79
Swollen or/and tender joints			
Shoulders	0.51	0.16–1.60	0.25
Elbows	0.93	0.30–2.94	0.91
Wrists	0.97	0.25–3.71	0.96
Fingers	0.83	0.10–7.13	0.86
Hips	1.32	0.27–6.49	0.73
Knees	0.63	0.20–1.97	0.42
Ankles	1.05	0.34–3.19	0.93
Toes	0.97	0.29–3.29	0.97
Patients with swollen large joints	0.59	0.19–1.79	0.35
Patients with swollen small joints	0.20	0.02-2.32	0.20
Number of swollen large joints	0.87	0.60–1.25	0.44
Number of swollen small joints	1.04	0.98–1.10	0.22
28 swollen joints	1.03	0.95–1.11	0.50
28 tender joints	1.004	0.92–1.10	0.92
Patients with erosion	1.63	0.51–5.21	0.41
Systemic signs and symptoms			
Temperature ≥ 38 °C	8.20 × 10^–7^	0–>10^6^	0.99
Malaise or fatigue	2.31	0.45–11.97	0.32
Weight loss	0.57	0.07–4.63	0.60
Morning stiffness (lasting at least 1 h)	1.45	0.42–4.96	0.56
Edema (both hands and feet)	4.83	1.50–15.56	0.034
Edema (only hands)	6.45 × 10^–7^	0–>10^5^	0.99
Edema (only feet)	2.78 × 10^–7^	0–>10^6^	0.99
CRP	1.08	1.18–0.92	0.08
ESR			
Men+Women	0.999905	0.98–1.02	0.08
Men	1.006	0.988–1.02	0.51
Women	0.98	0.95–1.01	0.25
Alb	0.63	0.24–1.65	0.35
LDH	1.0009	0.99–1.02	0.90
MMP–3			
Men +Women	1.00009	0.9992–1.001	0.84
Men	1.0006	0.9993–1.002	0.34
Women	0.9985	0.99–1.003	0.50
Hb			
Male + Women	0.77	0.57–1.06	0.11
Men	0.51	0.33–0.81	0.005
Women	1.47	0.80–2.71	0.21
Patients with RS3PE	4.83	1.50–15.56	0.034
Patients with seronegative RA	0.21	0.06–0.07	0.034
Patients fulfilling the classification criteria for RA [11,12]	0.27	0.07–0.98	0.046
Patients fulfilling the classification criteria for PMR [10]	0.50	0.06–4.02	0.51
Patients fulfilling the classification criteria for RA [11,12] + PMR [10]	2.82 × 10^–7^	0–>10^6^	0.99

Alb, albumin; CRP, C-reactive protein; ESR, erythrocyte sedimentation rate; Hb, hemoglobin; IQR, interquartile range; LDH, lactate dehydrogenase; MMP-3, matrix metalloproteinase 3; PMR, polymyalgia rheumatica; RA, rheumatoid arthritis; RS3PE, remitting seronegative symmetrical synovitis with pitting edema; SD, standard deviation.

## Data Availability

Not available.

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
