# Peer review of "Comparing the Clinical and Laboratory Features of Remitting Seronegative Symmetrical Synovitis with Pitting Edema and Seronegative Rheumatoid Arthritis"

_jcm, 2021, doi:10.3390/jcm10051116_

Round 1
Reviewer 1 Report
Dear authors,
Thank you for your time and effort to prepare the manuscript. The article has significant clinical relevance, although it could be improved in several instances please see my suggestions below:
1.Table 1: I suggest to provide this table in the supplementary materials and replace it with a table specifying the descriptive statistic ie. mean, n (%)
2. Figure 1 A: I did not find figure 1 A informative. it could be removed or palced in the supplementary materials.
3. statistics: Please report all significant results with 3 decimals and non significant results with two decimals in all tables for consistency.
Besides, please add how you treated missing values in the study.
4. Table 7: please provide statistic results for table 7, and the overall significance for the model you have reported. Please add to statistic section how you built the regression model. Is this based on the clinical relevance? or based on the all available data.
5. at last, I suggest to report most important results in the main text and place other results in the supplementary materials.
Sincerely yours
Author Response
Thank you very much for your comments.
Point 1:
Table 1: I suggest to provide this table in the supplementary materials and replace it with a table specifying the descriptive statistic ie. mean, n (%)
Response 1:
I agree with you.
Table 1 was moved to supplementary materials. The descriptive statistic of the 24 patients with RS3PE and 124 patients with seronegative RA was shown in Table 1.
Point 2:
Figure 1 A: I did not find figure 1 A informative. it could be removed or palced in the supplementary materials.
Response 2:
Figure 1 A was moved to supplementary materials.
Point 3:
statistics: Please report all significant results with 3 decimals and non significant results with two decimals in all tables for consistency.
Besides, please add how you treated missing values in the study.
Response 3:
We agree with you. In the case of missing data, the number of patient with available data was specified. We added the following in the manuscript: In the case of missing data, the number of patient with available data was specified(p.3, l.120). We showed the number of patient with available data in Tables.
Point 4: Table 7: please provide statistic results for table 7, and the overall significance for the model you have reported. Please add to statistic section how you built the regression model. Is this based on the clinical relevance? or based on the all available data.
Response 4:
This is not the result of multiple logistic regression analysis, but of univariate analysis. There were 14 patients with malignancy. Only univariate analysis was possible because of small sample size.
The following has been added to the manuscript (p. 3, l. 116-120)(p. 8, l. 208).
- 3, l. 116-120: The odds ratio(OR) and its 95% confidence interval (95% CI) indicate the increased or decreased risk of malignancy associated with a one-unit change in the predictor variable for continuous variables. For dichotomous variables, the OR indicate the risk of malignancy associated with the presence of the characteristic compared to the absence of the characteristic.
- 8, l. 208: Table 5. Risk factors for malignancy in patients with RS3PE or seronegative RA analyzed by univariate logistic regression analysis
Point 5: at last, I suggest to report most important results in the main text and place other results in the supplementary materials.
Response 5:
I agree with you.
Table 1, 2, 8, and 9 were moved to supplementary materials.
Reviewer 2 Report
The authors investigated the clinical difference between RS3PE and seronegative RA. This manuscript is valuable because these two disease sometimes show similar clinical manifestation and difficult to different diagnosis.
The number of RS3PE patients were small, but the conclusion provides useful information for clinicians. Moreover, the discussion and methodology were written appropriately.
Author Response
Comments:
The authors investigated the clinical difference between RS3PE and seronegative RA. This manuscript is valuable because these two disease sometimes show similar clinical manifestation and difficult to different diagnosis.
The number of RS3PE patients were small, but the conclusion provides useful information for clinicians. Moreover, the discussion and methodology were written appropriately.
Response:
Thank you very much for your comments. We are thankful for the time and energy you expended.
Round 2
Reviewer 1 Report
Dear authors,
Thanks for the revised version. It improved significantly. I have no further comments.
Best